# Toughening of Ni-Mn-Based Polycrystalline Ferromagnetic Shape Memory Alloys

**DOI:** 10.3390/ma16165725

**Published:** 2023-08-21

**Authors:** Siyao Ma, Xuexi Zhang, Guangping Zheng, Mingfang Qian, Lin Geng

**Affiliations:** 1School of Materials Science and Engineering, Harbin Institute of Technology, Harbin 150001, China; 2Department of Mechanical Engineering, The Hong Kong Polytechnic University, Hong Kong, China

**Keywords:** solid-state refrigeration, elastocaloric effect, ferromagnetic shape memory alloy, toughening, mechanical properties

## Abstract

Solid-state refrigeration technology is expected to replace conventional gas compression refrigeration technology because it is environmentally friendly and highly efficient. Among various solid-state magnetocaloric materials, Ni-Mn-based ferromagnetic shape memory alloys (SMAs) have attracted widespread attention due to their multifunctional properties, such as their magnetocaloric effect, elastocaloric effect, barocaloric effect, magnetoresistance, magnetic field-induced strain, etc. Recently, a series of in-depth studies on the thermal effects of Ni-Mn-based magnetic SMAs have been carried out, and numerous research results have been obtained. It has been found that poor toughness and cyclic stability greatly limit the practical application of magnetic SMAs in solid-state refrigeration. In this review, the influences of element doping, microstructure design, and the size effect on the strength and toughness of Ni-Mn-based ferromagnetic SMAs and their underlying mechanisms are systematically summarized. The pros and cons of different methods in enhancing the toughness of Ni-Mn-based SMAs are compared, and the unresolved issues are analyzed. The main research directions of Ni-Mn-based ferromagnetic SMAs are proposed and discussed, which are of scientific and technological significance and could promote the application of Ni-Mn-based ferromagnetic SMAs in various fields.

## 1. Introduction

Global warming and rapidly increasing energy consumption are becoming prominent issues in human society. The replacement of conventional gas compression refrigeration technology with novel refrigeration technologies that are environmentally friendly and efficient in energy conversion has attracted much attention. Solid-state refrigeration technology utilizes the reversible endothermic and exothermic effects of materials under the application of external fields, such as stress and magnetic fields, in refrigeration cycles, and has the advantages of small size and high efficiency for energy conversion. In addition, it does not emit working fluids or gases that deplete ozone and cause global warming, and it could be more environmentally friendly. Therefore, solid-state refrigeration technology has developed rapidly in the past decade [1,2,3,4,5,6].

The application of solid-state refrigeration technologies depends on the development of high-performance refrigerants [1,7,8,9]. At present, elastocaloric refrigeration based on the elastocaloric effect (eCE) of shape memory alloys (SMAs) and magnetocaloric refrigeration based on the magnetocaloric effect (MCE) are considered the most promising solid-state refrigeration technologies. Particularly, ferromagnetic SMAs have been developed for magnetocaloric refrigeration with high performance. Among these ferromagnetic SMAs, Ni-Mn-based alloys exhibit both first-order martensitic structural transformation (a cubic-to-tetragonal martensitic phase transformation) and second-order magnetic transformation. They could generate a magnetocaloric effect under applied magnetic fields, which are accompanied by the elastocaloric or piezocaloric effect under stress fields, as caused by the structural phase transition [10]. The multi-caloric effects that occur in Ni-Mn-based SMAs are very attractive in the practical applications of solid-state refrigeration.

The elastocaloric effect is generated from reversible super-elastic deformation under applied stress, which requires elastocaloric refrigerants to have good toughness and fatigue resistance. However, Ni-Mn-based alloys tend to show intergranular fracture due to their elastic anisotropy. They are prone to structural fatigue (e.g., the initiation and propagation of cracks) and functional decay fatigue (e.g., the degradation of elastic thermal effects and reduced reversibility) [1,11]. In the past decade, there have been many studies on toughening methods and toughening mechanisms for Ni-Mn-based alloys, but a systematic summary of the progress in these studies is lacking. In this review, the recent advances in the development of ferromagnetic SMAs with multi-caloric effects are summarized, with a focus on identifying the mechanisms that could cause the low intrinsic toughness of Ni-Mn-based alloys. Then, the effects of element doping, microstructure, size effect, and *d*-orbital hybridization on the toughness, martensitic transformation, and elastocaloric effect of Ni-Mn-based alloys are systematically summarized. Finally, the research directions and prospects are given, providing guidelines for improving the toughness of Ni-Mn-based alloys.

## 2. Crystal Structure and Mechanical Properties of Ni-Mn-Based Alloys

In 1996, Ullakko et al. [12] first reported the magnetically induced strain of single-crystal Ni-Mn-Ga alloys, which aroused great interest in ferromagnetic shape memory alloys (FSMAs). The Ni-Mn-Ga alloy is a Heusler alloy, which is one of the ordered intermetallic compounds with the space group $Fm\bar{3}m$. The stoichiometric formula of such an alloy can be expressed as X_2_YZ, where X is a transition metal element, such as Ni, Fe, Co, etc., or Au, Cu, Pd, and other precious metal elements; Y is a transition metal or rare earth metal element; and Z can be an *s-p* metal element, such as In, Ga, Al, Sn, Sb, etc. Taking the Ni-Mn-Ga alloy as an example, the crystal structure of X_2_YZ alloys can be considered to be composed of four face-centered sub-lattices interspersed with each other along the diagonal direction [13]. The constituent elements of these four sub-lattices are Ga, Ni, Mn, and Ni, which have fractional coordinates of (0, 0, 0), (1/4, 1/4, 1/4), (1/2, 1/2, 1/2), and (3/4, 3/4, 3/4), respectively. It has been found that martensitic transformations can occur in Ni-Mn-X alloys (X = Ga, Sn, In, Sb) with non-stoichiometric lattice structures [14,15]. During martensitic transformation, the parent phase can be transformed into martensite with different modulation structures, e.g., Ni-Mn-Ga alloys can be transformed into martensite with a system-dependent symmetry-modulated structure (non-monoclinic, monoclinic 10 M, or monoclinic 14 M structures) [16].

Ni-Mn-X (X = Ga, In, Sn, Sb) alloys are prone to intergranular fracture, and their brittle fracture originates from intrinsic brittleness and environmental brittleness. Intrinsic brittleness is caused by the large differences in the atomic size and electronic structure of the Ni, Mn, and X elements. From the point of view of electronic structures, Ni (3d^8^4s^2^) and Mn (3d^5^4s^2^) are transition metal elements, and their second outer d orbital is the valence orbital. On the contrary, Ga (3d^10^4s^2^4p^1^), In (4d^10^5s^2^5p^1^), Sn (4d^10^5s^2^5p^2^), and Sb (4d^10^5s^2^5p^3^) are the main group *s-p* elements, and their outermost p orbital is the valence orbital. Ni-Mn-based alloys are relatively brittle due to the weak p-d covalent hybridization between the sub-outer and outermost orbitals [17,18,19]. In addition, polycrystalline Ni-Mn-X alloys have strong elastic anisotropy, and stress concentration easily occurs at grain boundaries. Furthermore, Ni-Mn-X alloys have an ordered structure with a large volume of unit cells and a large Burgers vector of dislocations, leading to difficulty in dislocation slips and a reduction in independent slip systems, which could result in the intrinsic brittleness of the alloy [17,18,19].

Environmental brittleness originates from the chemical interactions between grain boundary atoms and external substances. The formation of chemical bonds between grain boundary atoms and impurity atoms leads to a reduction in the atomic binding force at the grain boundary. Environmental brittleness can be alleviated by reducing grain boundary areas in polycrystalline alloys or avoided by using single-crystal alloys. Czaja et al. [20] reported that single-crystal Ni_49.5_Mn_38.4_Sn_12.2_ alloys could give rise to 4.9% recoverable strain, and strain of up to 9.7% could be obtained under applied stress of 600 MPa. Chernenko et al. [21] reported a recoverable strain as high as 12.0% for <001> oriented Ni_49_Mn_28_Ga_23_ single crystals. Wang et al. [22] found that the Ni_57_Mn_18_Ga_21_In_4_ single crystal produced a recoverable strain of up to 7% under uniaxial compressive stress of 150 MPa. In general, the mechanical properties of the aforementioned single-crystal alloys are better than those of polycrystalline alloys. Since the elastocaloric effect originates from the stress-induced reversible transformation of martensite between the parent phase and the martensite [10], the stress concentration at the grain boundary is likely to occur during phase transformation, leading to crack initiation and failure [23], which could seriously restrict the application of Ni-Mn-based alloys [3,24].

On the basis of the mechanisms of brittleness that could exist in Ni-Mn-based ferromagnetic SMAs, viable approaches to improving the plasticity and toughness of Ni-Mn-based alloys are proposed, which mainly include: (1) the improvement of the mechanical properties of the parent phase through element doping, solid solution strengthening, or introducing a ductile second phase; (2) the purification and modification of the grain boundary by element doping, which could improve the bonding forces at the grain boundary and reduce the tendency of intergranular fracture; (3) the design of microstructure or the adjustment of grain size and crystal orientation (texture); (4) the replacement of main group elements (Ga, In, Sn, Sb) by transition metal *3d* elements with *d-d* orbital hybridization that could reduce brittleness.

## 3. The Effects of Doping on the Toughness of Ni-Mn-Based Alloys

### 3.1. The Solid Solution and Second-Phase Toughening Effects Resulting from Doping of Transition Metal Elements

Transition metal elements, such as Cu, Fe, Co, Cr, etc., can be doped into Ni-Mn-based alloys, resulting in either a solid solution or a second-phase toughening effect. Doping Cu into Ni-Mn-Ga alloys promotes the coupling of martensitic transformation and magnetic transformation, which can generate s giant magnetocaloric effect and a large magnetically induced strain [25,26]. Meanwhile, Cu doping can improve the plasticity of Ni-Mn-based alloys and reduce the cost of materials. In 2010, Wang et al. [27] pointed out that the Ni_50_Mn_25_Ga_17_Cu_8_ alloy has a single-phase structure at room temperature, and its compressive strength and fracture strain are 878 MPa and 22%, respectively. Subsequently, Wang et al. [28] reported a more ductile dual-phase Ni_30_Cu_20_Mn_41.5_Ga_8.5_ alloy with a compressive fracture strain exceeding 70%. In situ observations during dynamic stretching indicated that the second phase distributed along the grain boundaries is the key to the enhancement of the plasticity of the alloy. Li et al. [29] used Cu to replace Mn in Ni-Mn-Sn alloys and found that the fracture strength of the Ni_44_Mn_43_Sn_11_Cu_2_ alloy with a dual-phase structure was as high as 1150 MPa. The elastic modulus could reach 400 GPa, which was twice that of the single-phase Ni-Mn-Sn-Cu alloy.

It has been reported that the doping of a certain amount of the ferromagnetic element Fe into Ni-Mn-Ga alloys can improve the magnetic properties of the alloy; thus, the doping of the Fe element, which might be effective in enhancing the mechanical properties of Ni-Mn-based magnetic SMAs, has attracted attention [30]. Similar to Cu doping, Fe doping can also improve the plasticity of the alloy, but Fe doping more likely results in the formation of a second phase. In 2001, Cherechukin et al. [31] found that polycrystalline Ni-Mn-Ga alloy doped with a small amount of Fe could improve the plasticity of the alloy without affecting its magnetocaloric properties. Later, Feng et al. [30] replaced Mn with Fe in the Ni-Mn-Ga alloy; the as-prepared Ni_52_Mn_9_Fe_15_Ga_24_ single crystal had significantly improved plasticity, and its Young’s modulus along the [001] direction and Vickers hardness reached 13.7 GPa and 6.4 GPa, respectively. Wang et al. [32] systematically studied the effects of Fe content on the hardness and toughness of the Ni_48.7_Mn_30.1−x_Fe_x_Ga_21.2_ alloy and found that the hardness and toughness increased after Fe doping. Studies on the fracture of alloys indicated that the Ni_48.7_Mn_30.1_Ga_21.2_ (x = 0) alloy had an intergranular fracture with a fracture toughness of 12 N/mm^3/2^; while the Ni_48.7_Mn_19.1_Fe_11_Ga_21.2_ (x = 11) alloy possessed a transgranular fracture with a fracture toughness of 18 N/mm^3/2^. The results indicated that Fe doping could strengthen the grain boundaries and suppress the tendency of intergranular fracture. Feng et al. [33] found that with an increase in Fe content, the compressive strength and fracture strain of the Ni_50_Mn_34_In_16−y_Fe_y_ alloy increased, and a transition from intergranular to transgranular fractures occurred; when the Fe content was more than 5 at.%, the second phase was precipitated at the grain boundaries and inside the grain interiors; when the Fe content was 8 at.%, the compressive strength and maximum compressive strain reached 1200 MPa and 15.8%, respectively. Recently, Pfeuffer et al. [34] proposed a method to design the microstructure of Ni-Mn-In-based alloys through Fe doping. It was found that the Fe-enriched and In-depleted secondary phases were precipitated and connected at the grain boundaries, resulting in excellent mechanical stability, and the intergranular fracture during cyclic loading was hindered. In this way, the alloy withstood more than 16,000 mechanical cycles under external stress of up to 300 MPa without structural degradation.

With cobalt doping, Ni-Mn-based alloys can possess a giant magnetocaloric effect, giant magnetoresistance, magnetic superelasticity, and a magnetic shape memory effect [35,36,37,38,39], as well as improved plasticity. Therefore, Co doping is widely used in the modification of Ni-Mn-X (X = Ga, Sn, In, Sb) alloys. In 2001, Oikawa et al. obtained a γ-phase in Ni-Fe-Ga [40] and Co-Ni-Al [41] ferromagnetic alloys, which improved the toughness of the alloys; the doping of the Co element is also beneficial to the formation of γ phase in the matrix and the improvement of alloy toughness. Ma et al. [42] found that the ductility and hot workability of Ni-Mn-Co-Ga alloys improved with increasing content of γ phase, and the tensile strength and fracture strain of Ni_56_Mn_17_Co_8_Ga_19_ alloy reached 729.3 MPa and 14.1%, respectively. Dynamic tensile tests and fracture observations confirmed that the microcracks stopped growing when they reached the boundary between the martensite and the γ phase, confirming that the γ phase could hinder crack growth. Shen et al. [43] studied the microstructure and mechanical properties of Ni_45_Mn_37−x_In_13_Co_5_Cr_x_ (x = 0, 1 and 2) polycrystalline alloys and found that a ductile second phase was formed in the alloy after Cr doping, which resulted in a fracture strain of Ni_45_Mn_36_In_13_Co_5_Cr alloy as high as 5%.

The ductile second phase formed after doping can improve the plasticity of Ni-Mn-based alloys. However, the second phase hinders the martensitic transformation and increases the transformation temperature hysteresis. The reason is that during the phase transition, the interface between the matrix and the second phase has lattice distortion, which generates dislocations and hinders the migration of the phase interface. As a result, the phase transition temperature hysteresis increases.

### 3.2. Grain Refinement and Second Phase Toughening Effects of Rare Earth Element Doping

In 2004, Li et al. [44] obtained a Ni_54_Mn_25_Ga_21_ alloy with a grain size of about 10–50 μm through rapid solidification. The compressive strength and maximum compressive strain of the alloy were 970 MPa and 16%, respectively, while the compressive strength and strain of the alloy with a grain size of 200 μm were only 440 MPa and 10%, respectively. Rare earth elements can refine the grains during the melting process, thereby effectively improving the mechanical properties of Ni-Mn-based alloys. Zhang et al. [45,46] found that in the Ni_54_Mn_25_Ga_21−x_Gd_x_ alloy, when the Gd content was less than 0.1 at.%, the alloy had a single-phase structure; when the Gd content was 1%, the alloy could have a precipitated Gd-rich phase. The compressive strength and strain of Ni_54_Mn_25_Ga_20_Gd_1_ alloy are 958 MPa and 16%, respectively. The doping content of rare earth elements determines the strengthening mechanism in the alloy. When the content of Gd is not compatible with that for the formation of a second phase, the strengthening mechanisms are mainly fine-grain strengthening and solid solution strengthening. Dong et al. [47] increased the amount of Gd doping in the alloy and studied the toughening effect of the Gd-rich second phase. It was found that the compressive strength of the Ni_45.4_Mn_39.5_In_13.1_Gd_2_ alloy was 2065 MPa, and the fracture strain was 9.32%. Compared with that of the Ni_45.4_Mn_41.5_In_13.1_ alloy without the second phase, the compressive strength was increased to 1700 MPa. Tan et al. [48] found that a Gd-rich second phase appeared in Ni-Mn-Sn-Gd alloys when the content of Gd doping exceeded 0.5 at.%. When the Gd content was increased from 0 to 2 at.%, the compressive strength increased from 448 MPa to 707 MPa, and the fracture strain increased from 4.5% to 9.0%; in addition, with increasing Gd content, the transition from intergranular fracture to transgranular fracture in the alloy could occur. Li et al. [49] found that when the Gd content was 1 at.%, the compressive strain of the Ni_45_Co_5_Mn_37_In_12_Gd_1_ alloy reached the maximum (8.8%); when the Gd content was 2 at.%, the compressive strength was the highest (992 MPa), which was 5.5 times that of the alloy without Gd doping, and the maximum compressive strain was slightly lower than that of the alloy with 1 at.% Gd doping, as shown in Figure 1a. Compared with Ni, Mn, Ga, and In, Gd has a much larger atomic radius. Therefore, most of the doped Gd atoms are distributed at the grain boundaries, and the formation of the second phase drags the grain boundaries and hinders the growth of grains. As a result, the excessive doping of Gd leads to a reduction in plastic strain, which is mainly attributed to the aggregation and growth of an excessive second phase. Therefore, the mechanical properties of the alloy are reduced.

Similar to Gd doping, Tb doping also refines the grains. Nonetheless, excessive Tb doping has obvious negative effects on the mechanical properties of the alloy. Since the Tb-rich second phase is more brittle, during the deformation process, fracture occurs earlier in the Tb-rich second phase compared with that in the matrix, resulting in a decrease in the mechanical properties of the alloy. Tian et al. [50] added Tb into Ni-Mn-Sn alloys and found that the compressive strength of the Ni_48_Mn_39_Sn_11_Tb_2_ alloy was 571.8 MPa and the fracture strain was 22.0%; the compressive strength of the Ni_48_Mn_39_Sn_13_ alloy was only 74.3 MPa, and the fracture strain was 9.2%. However, when the Tb doping content exceeded 2 at.%, the mechanical properties of the alloy decreased with increasing Tb content, as shown in Figure 1c. Shen et al. [51] investigated the effect of Tb doping on the mechanical properties of Ni_50_Mn_34_In_16−x_Tb_x_ (x = 0, 0.1, 0.3, 0.4, 0.5) alloys. When x < 0.3, the alloys possessed a single phase. When x > 0.3, the Tb-rich second phase was distributed along the grain boundary of the parent phase, the fracture mode changed from intergranular fracture to mixed intergranular–transgranular fracture and the plasticity improved. In addition, the average grain size of the alloy was about 50 μm for x = 0.3 and about 5 μm for x = 0.4. Wu et al. [52] studied the microstructure and mechanical properties of the Ni_50−x_Tb_x_Mn_30_Ga_20_ (x = 0.1–1) alloy and found that with increasing content of the second phase, the compressive strength and strain increased gradually when 0.1 < x < 0.2; when 0.2 < x < 1, they gradually decreased, and the fracture mode changed from intergranular fracture mode to transgranular fracture mode. Lee et al. [53] found that the compressive strength of the Ni_48.8_Mn_29.7_Ga_21.5_ alloy doped with 0.3 at.% Tb was 780 MPa, and the fracture strain was 0.078. Figure 1b shows the relationship between Tb doping and the mechanical properties of the alloy. It is believed that the improvement of mechanical properties by the addition of Tb is related to grain refinement and grain boundary strengthening. It was found that excessive Tb doping causes a decrease in alloy toughness, and the optimum doping amount of Tb in different alloys could vary. In particular, alloys with 0.2–0.4 at.% Tb doping have the most significant improvement in their mechanical properties.

The effects of other rare earth metals, such as Nd, Dy, and Y, on the plasticity of Ni-Mn-based alloys have also attracted much attention. Tsuchiya et al. [54] studied the effect of Nd on the mechanical properties of Ni-Mn-Ga alloys and found that alloys with 0.3 at.% Nd doping had a compressive strength and fracture strain of about 700 MPa and 6%, respectively; in comparison, the compressive strength was only 200 MPa and the fracture strain was 2% in the alloy without Nd doping. Figure 1d shows the effects of Nd doping on the mechanical properties of the alloy, suggesting that the improved plasticity in the alloys with Nd doping could result from solid solution strengthening and grain boundary purification. Gao et al. [55,56,57] systematically studied the effects of Dy on the microstructure and mechanical properties of Ni_50_Mn_29_Ga_21−x_Dy_x_ (0 ≤ x ≤ 5) alloys, as shown in Figure 1e. It was found that the alloy was in a single phase when x < 0.5, and the second phase of Dy(Ni,Mn)_4_Ga with a hexagonal CaCu_5_ structure appeared when x ≥ 0.5. When the Dy content increased from 0.5 at.% to 2 at.%, the Dy(Ni,Mn)_4_Ga phase was distributed along the grain boundaries, and the matrix and Dy(Ni,Mn)_4_Ga phase formed a eutectic structure. When the Dy content was less than 1 at.%, the compressive strength and fracture strain increased significantly with increasing Dy content; the fracture strain reached the maximum in alloys with 1 at.% Dy doping. With increasing Dy content, the fracture characteristics of the alloy changed from intergranular fractures to transgranular cleavage fractures. Recently, Tong et al. [58] showed that the addition of a small amount of Dy could also improve the toughness of the alloy. Through microalloying with Dy, DyNi_4_Ga granular precipitates were formed in the matrix, and the compressive strength and fracture strain of the Ni_54_Mn_25_Ga_20.9_Dy_0.1_ alloy at room temperature were ~855.9 MPa and ~12.7%, respectively.

Cai et al. [59] studied the effect of Y doping on the microstructure and mechanical properties of Ni-Mn-Ga alloys. The addition of Y increased the compressive strength of the Ni_50_Mn_28_Ga_22−x_Y_x_ alloy, and the compressive strength of the Ni_50_Mn_28_Ga_19_Y_3_ alloy reached 1279 MPa, which was about 900 MPa higher than that of the alloy without Y doping; with increasing content of Y, the fracture strain increased gradually and reached a maximum value of 16% in the alloy with 1 at.% Y doping. However, the fracture strain decreased slightly when the Y content was greater than 1 at.%, which could have been caused by the network-like distribution and local enrichment of the Y-rich second phase. The work conducted by Sui et al. [60] showed that the microstructure of Ni-Mn-Ga-Y alloy was composed of matrix and Y-rich second phase, and the Y-rich Y(Ni,Mn)_4_Ga phase had a hexagonal CaCu_5_ type crystal structure evenly distributed inside the grains. With increasing Y doping, the Y-rich second phase segregated at the grain boundary. Meanwhile, the Y element refined the grains in the alloy. As shown in Figure 1f, Y doping improved the yield strength and plasticity of the alloy, and the fracture type changed from intergranular fracture to transgranular cleavage fracture.

**Figure 1 materials-16-05725-f001:**
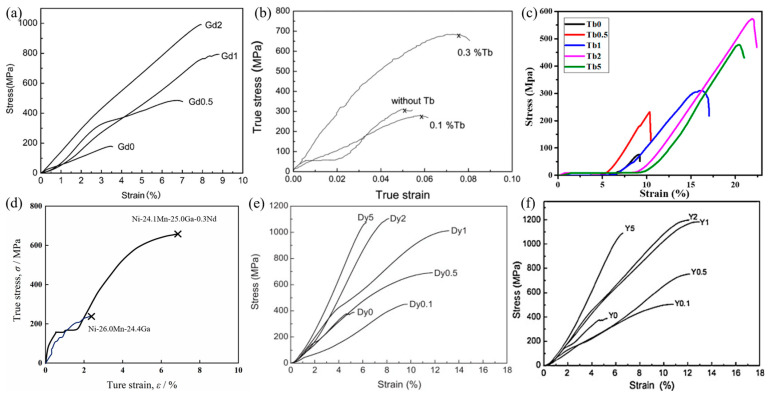
Toughening effect of rare earth elements in Ni-Mn-based shape memory alloys. (**a**) The compressive stress–strain curves of Ni_45_Co_5_Mn_37_In_13−x_Gd_x_ alloys at room temperature (reproduced with permission from [49], Elsevier, 2018); (**b**) the compressive stress–strain curves of Ni-Mn-Ga alloys and Tb-doped alloys (reproduced with permission from [53], Springer Nature, 2009); (**c**) the compressive stress–strain curves of Ni_48_Mn_39_Sn_13−x_Tb_x_ alloys [50], reprinted under a Creative Commons Attribution (CCBY) license; (**d**) the compressive stress–strain curves of ternary Ni-Mn-Ga and Ni-Mn-Ga-Nd alloys (reproduced with permission from [54], Elsevier, 2004); (**e**) the compressive stress–strain curves of Ni_50_Mn_29_Ga_21−x_Dy_x_ alloys at room temperature (reproduced with permission from [57], Elsevier, 2012); (**f**) The compressive stress–strain curves of Ni_50_Mn_29_Ga_21−x_Y_x_ alloys at room temperature (reproduced with permission from [60], Elsevier, 2011).

According to the research results mentioned above, the doping of rare earth elements has an obvious effect on improving the mechanical properties of Ni-Mn-based alloys. The mechanisms by which the plasticity of Ni-Mn-based alloys is improved by doping an appropriate amount of rare earth elements include grain refinement, the introduction of ductile second phases, and the purification of grain boundaries. This is because rare earth elements can react with some harmful impurities (such as O, S, Pb, etc.) to inhibit the segregation of impurity elements at grain boundaries; in addition, the rare earth metals can gather at the grain boundaries and become obstacles to the growth of grains, thereby refining the grains. For alloys with doping of rare earth elements, it has been found that Gd doping makes the grain refinement of the alloys more obvious, and the mechanical strength and fracture strain of alloys doped with Gd are higher than those of alloys doped with other rare earth elements. 

However, at present, the research on the optimal doping amount of rare earth elements is not systematic. The synergistic effect of multiple rare earth element co-doping or mixed doping of rare earth and other metal elements needs further study. In doped alloys, the resulting second phases have different mechanical properties, and the toughening effects of different rare earth elements are very different. A rare earth metal-rich second phase with an appropriate content could be distributed along the grain boundary, effectively hindering grain growth and preventing the initiation and propagation of cracks. However, if the brittle rare-earth-rich second phase is excessive, it could have a network-like distribution along the grain boundary or be enriched locally, breaking up the continuity of the matrix so that cracks are easily initiated at the grain boundary or phase interface. As a result, the toughness of the alloy is reduced. Furthermore, the matrix and the second phase may form a eutectic structure, and the flaky second phase in the eutectic cells reduces the plasticity of the alloy. Therefore, it is necessary to adjust the solidification conditions and microstructure of the alloys.

### 3.3. Grain Refinement and Second Phase Toughening Effects of Metalloid Element Doping

#### 3.3.1. Second Phase Toughening by Metalloid Element Doping

The ductility of Ni-Mn-based alloys can be improved by doping the B element, introducing second-phase particles at grain boundaries. In 2011, Nong et al. [61] studied Ni-Mn-Sb alloys with B doping and found that if the content of the B element was increased from 1 at.% to 3 at.%, the Curie temperature increased from 330 K to 345 K, and the martensitic transformation temperature decreased from 300 K to 262 K. Prusik et al. [62] investigated the Ni-Mn-Co-In alloy doped with the B element and found that the B element facilitated the formation of “Co-rich and In-deficient” γ phase and M_23_B_6_ phase. When the B content reached 1 at.%, the second phases formed a “sub grain “ structure within the grain interiors and at the grain boundaries. Aydogdu et al. [63] added the B element to Ni-Mn-Ga alloys and found that the compressive strength and plasticity of the alloys were improved. When the content B doping was 1 at.%, the compressive strength reached 1100 MPa, and the fracture strain reached 30%. The compressive strength and fracture strain of the undoped Ni_51_Mn_28.5_Ga_19.5_ alloy were only 300 MPa and 11.5%, respectively, as shown in Figure 2f. However, when the content of B doping increased from 1% to 3%, the yield strength of the alloy increased, and the ductility decreased. Zhang et al. [64] found that by doping B, the grains of the (Ni_54_Mn_25_Ga_21_)_100−x_B_x_ alloy were significantly refined. When the content of B doping was 3 at.%, the grain size decreased from several hundred microns in undoped alloys to 20 microns in the doped ones, and the compressive strength of the alloys with B doping first increased and then decreased with increased content of B doping. When the content of B doping was 1 at. %, the compressive strength of the alloy was the highest (1100 MPa), which was about 700 MPa higher than that of the undoped Ni_54_Mn_25_Ga_21_ alloy. Figure 2g shows the effect of B doping on the mechanical properties of the alloys.

There are few studies on the effects of Al doping on the microstructure and mechanical properties of Ni-Mn-based alloys. Barman et al. [65] studied the mechanical properties of Al-doped Ni-Mn-Sb-Al thin films and found that with an increasing Al content of doping, the grain size decreased, and compared with the undoped Ni_50.3_Mn_36.9_Sb_12.8_ alloy, Ni_49.7_Mn_36.4_Sb_8.3_Al_5.6_ films had higher hardness (12.6 ± 2.2 GPa) and elastic modulus (280 ± 2.3 GPa).

#### 3.3.2. Grain Boundary Purification and Modification due to Metalloid Element Doping

According to the microalloying theory, which is different from the second phase strengthening and toughening mechanisms, Yang et al. proposed that microalloying could enhance the grain boundary strength and refine the grains, thereby improving the mechanical properties and cycling stability of alloys with B doping [66,67]. The B element microalloying theory can be applied to Ni-Mn-In-Fe quaternary alloys [67], with a conceptual diagram shown in Figure 2a. The toughening effects of B microalloying on the structural properties of (Ni_51.5_Mn_33_In_15.5_)_100−x_B_x_ (x = 0, 0.1, 0.2, 0.3, 0.4, 0.6) alloys are shown in Figure 2b–d. B microalloying influences the mechanical properties of the alloys as well, which are shown in Figure 2e–g. For example, the Ni_51.5_Mn_33_In_15.5_ alloy failed after 20 stress cycles, while the (Ni_51.5_Mn_33_In_15.5_)_9.7_B_0.3_ alloy remained basically stable after more than 150 stress cycles, as shown in Figure 2e. Tang et al. [4] studied the Ni-Mn-In alloy with co-doped Cu and B elements. With increasing content of Cu doping, the Ni-rich second phase was precipitated in the matrix, which improved the mechanical properties of the alloy; during elastocaloric tests on the (Ni_52_Mn_31_In_16_Cu_1_)_99.8_B_0.2_ polycrystalline alloy, an adiabatic temperature change of 9.5 K was obtained under applied stresses of up to 220 MPa. After 100 cycles, the adiabatic temperature change became stable at 6.8 K with a high strain of 3%. The results indicate that stability in the plasticity and elasticity of the alloy were improved after Cu and B co-doping.

It is worth noting that although the doping amount of B element in Ni-Mn-based alloys can be as high as 5 at.%, the content of microalloying generally does not exceed 0.6 at.%. This can be explained by the differences in the toughening mechanisms related to B element microalloying and the formation of second phase, as follows: the microalloying of the B element promotes the formation of the coordination compound NiBH^+^, and NiBH^+^ exhibits strong grain boundary segregation. Hydrogen is bonded to B and Ni and trapped in the NiBH clusters, which effectively reduces the diffusion of hydrogen along grain boundaries and suppresses hydrogen embrittlement. Therefore, B element microalloying increases grain boundary strengthening, which inhibits the formation and propagation of cracks along the grain boundaries, thereby improving the toughness of the alloy.

**Figure 2 materials-16-05725-f002:**
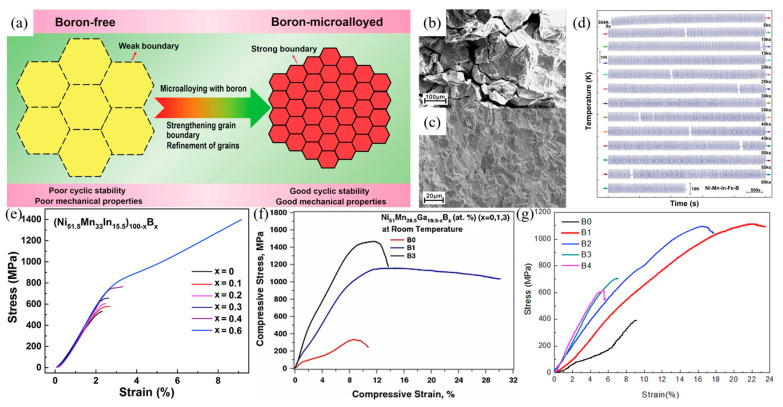
Toughening effect of boron in Ni-Mn-based shape memory alloy. (**a**) Schematic illustration of boron microalloying in enhancing the mechanical properties and cycle stability of shape memory alloys [67], reprinted under a Creative Commons Attribution (CCBY) license; scanning electron fractographs for (**b**) Ni_51.5_Mn_33_In_15.5_ and (**c**) (Ni_51.5_Mn_33_In_15.5_)_99.4_B_0.6_ [67], reprinted under a Creative Commons Attribution (CCBY) license; (**d**) temperature variation during cyclic loading, holding, and unloading of boron microalloying reinforced materials [67], reprinted under a Creative Commons Attribution (CCBY) license; (**e**) compressive stress–strain curves for (Ni_51.5_Mn_33_In_15.5_)_100−x_B_x_ (x = 0, 0.1, 0.2, 0.3, 0.4, 0.6) alloys, measured at A_f_+60 K (reproduced with permission from [66], Elsevier, 2017); (**f**) ductility behavior of Ni_51_Mn_28.5_Ga_19.5−x_B_x_ (x = 0, 1, 3) alloys under compression at room temperature (reproduced with permission from [63], Springer Nature, 2014); (**g**) compressive stress–strain curves of (Ni_54_Mn_25_Ga_21_)_100−x_B_x_ alloys (reproduced with permission from [64], Elsevier, 2016).

Cong et al. [68] developed new Ni-Mn-based magnetic SMAs with both low-field-driven magnetocaloric effects and excellent mechanical properties. By adding a small amount of Al to the Ni-Co-Mn-Sn alloy, the stacking fault energy could be reduced, resulting in a specific martensite accumulation-mediated structure (composed of fine martensite structures with different accumulation periods). The geometric compatibility of the martensite and parent phase was improved, and the thermal hysteresis of transformation was reduced. At room temperature, the Ni_40_Co_10_Mn_40_Sn_9_Al_1_ alloy had a compressive fracture strain of 8.7% and a compressive strength of 1050 MPa; while at a high temperature of 423 K, a large fracture strain of 5.6% and a high compressive strength of 690 MPa could still be obtained.

The doping-induced toughening methods, mechanisms, and effects for the aforementioned Ni-Mn-based SMAs are summarized in Table 1.

## 4. The Effects of Texture, Sizes, and *d*-Orbital Hybridization on the Toughness of Ni-Mn-Based Alloys 

### 4.1. The Effects of Texture on the Toughness of Ni-Mn-Based Alloys

According to the dislocation theory [69], the strength factor *K* of the Hall–Petch relation is *K* = *M*(*α·τ_CRSS_·G·b*)^1/2^, where *M* is the orientation factor of the polycrystalline structure; *τ_CRSS_* is the critical shear stress of a slip system; *G* is the shear modulus; *b* is the Burgers vector; *α* is a constant. Since the orientation factor is related to the texture structure of the polycrystalline Ni-Mn-based alloys, the texture structure can affect their strength factor *K* and mechanical properties [70].

The texture structure obtained by directional solidification technology can improve strain compatibility among grains, reduce stress concentration, and inhibit crack initiation, providing a viable method to reduce the brittleness of Ni-Mn-based alloys. Huang et al. [71,72] used directional solidification technology to prepare a columnar Ni-Mn-In alloy. The textured Ni_51.8_Mn_31.4_In_16.8_ alloys had a compressive strain as high as 7%, which was 1.7 times higher than that of the alloy with randomly oriented grains, and the compressive strength was as high as 700 MPa, which was twice that of the alloy with randomly oriented grains. Subsequently, Huang et al. [73] used the directional solidification technique to prepare a strongly textured Ni-Fe-Ga-Co polycrystalline alloy. When the loading direction was consistent with the grain boundary orientation (the solidification direction), the reversible superelastic strain was 5%. By contrast, only partial superelastic recovery could be observed in the alloy with a 90° loading direction. Li et al. [70] prepared a Ni_45.7_Co_4.2_Mn_37.3_Sn_12.8_ polycrystalline alloy with a preferred orientation along [553]_A_, which had a compressive strength of 740 MPa and a compressive strain as high as 3.7%; compared with the compressive strength of the Ni_46_Mn_38_Sb_12_Co_4_ polycrystalline alloy with disordered grain orientations (~100 MPa) [74], the mechanical properties of directionally solidified alloys have been greatly improved.

Zhao et al. [75] prepared a strongly textured Ni_50_Mn_31.5_In_16_Cu_2.5_ polycrystalline alloy and observed a superelastic strain of 3.5% and critical stress of about 130 MPa caused by phase transformation; in the non-textured polycrystalline alloys with the same composition, the slope of the superelastic plateau increased, indicating that the random distribution of grains is not favorable for the occurrence of superelasticity. In the Ni-Mn-In-Co polycrystalline alloy grown along the [001] crystallographic direction using the same preparation method, a superelastic plateau appeared at a critical stress of 100 MPa, and a total strain of 6.7% could be fully recovered after the stress was released [76]. The formation of texture not only improved the toughness of the alloy; Lu et al. [77] also found that the texture could reduce stress hysteresis. For example, the as-prepared Ni-Mn-In-Co alloy had a small stress hysteresis of about 34 MPa under a compressive strain of 6% along the [100]_A_ and [331]_A_ crystallographic directions.

Besides those prepared by directional solidification technology, textured Ni-Mn-based alloys prepared by other methods also exhibit a good toughening effect. Lu et al. [78] prepared a Ni_45.7_Mn_36.6_In_13.3_Co_5.1_ alloy by hot die casting. After two cycles of uniaxial compression along the [001] direction, the alloy did not fracture. The strength of the elastocaloric effect reached 35 K/GPa, which was higher than that of polycrystalline Cu-Zn-Al, Ni-Ti, and Fe-Pd alloys [79,80,81,82], while the undoped Ni-Mn-In alloys were too brittle to carry out the cycle tests. The textured polycrystalline Ni_50_Mn_32_In_16_Cr_2_ alloy prepared by Hernandez-Navarro et al. [83] by arc melting could withstand compressive stress of 100 MPa, and an adiabatic temperature change of 3.9 K could be obtained when stress was applied along the [001] direction.

Therefore, texture plays an important role in improving the mechanical properties of Ni-Mn-based alloys. After the directional solidification of Ni-Mn-based alloys, the strain compatibility of grain boundaries is improved, the stress concentration and crack initiation sensitivity are reduced, and the toughness is improved. More importantly, the texture structure introduced by directional solidification reduces the stress hysteresis and improves the cyclic stability and reversibility of superelasticity in an elastocaloric refrigeration cycle. At present, studies on the preparation of textured alloys by hot die casting and arc melting are relatively limited, and the effects of these methods on improving the toughness of Ni-Mn-based alloys are not as obvious as those of directional solidification technology.

In order to explain the effect of texture on the plasticity of Ni-Mn-based alloys, grain boundaries are assumed to be flat interfaces that play a role in stress transfer. Stress on the grain boundary can be divided into shear stress and compressive stress (Figure 3a), and the resolved compressive stress increases as the intersection angle (θ) increases. When the compressive stress at the grain boundary is lowered, the strain coordination of the adjacent grains is improved. When one grain transforms into its martensitic variant, each grain produces the same displacement on the grain boundary (Figure 3b). As a result, a symmetrical martensitic morphology is formed in the adjacent grains (Figure 3d), and a full superelastic recovery is obtained at an angle of 0°. On the contrary, the compressive stress is the largest when the intersection angle θ is 90°, resulting in stress concentration at the grain boundary and differences in the displacement of the adjacent grains (Figure 3c) and asymmetric morphology (Figure 3e). In this case, plastic deformation and cracks may occur near the grain boundaries (Figure 3e). Therefore, the introduction of texture is an effective method that could inhibit grain boundary cracking and improve the plasticity of alloys.

### 4.2. The Effects of Size on the Toughness of Ni-Mn-Based Alloys

A material’s characteristic sizes on the micro and nano scales, such as grain size and dimensions, can significantly affect its mechanical behavior [84,85]. Nanoindentation, tensile and torsion tests of fibers with diameters of microns, as well as bending and tensile tests of ribbons indicate that the yield strength of a material increases with a decreasing diameter or thickness of the samples. Therefore, when the size of a Ni-Mn-based alloy is on the micro or nano scale, the mechanical properties change greatly. 

In single-crystal Ni-Mn-Ga alloys, large strains can be induced by an applied magnetic field [20,21,22], which could be related to the low resistance of the motions of twin boundaries. However, single-crystal alloys are difficult and expensive to prepare. Polycrystalline alloys can be prepared rapidly and at a low cost by casting, but they have small magnetic field-induced strain due to the constraint effects of grain boundaries on the motion of the twin boundary. Dunand and Müllner [86] proposed that when the grain size of the Ni-Mn-Ga alloy is compatible with the characteristic sizes of the samples, such as the powder diameter, fiber diameter, film thickness, and the pore node diameter of porous materials, as shown in Figure 4a, the resistance of the motions of the twin boundary in the alloy can be reduced. In this way, there are more free surfaces around the grains, and the constraint effects of the grain boundaries are greatly reduced during the reorientation of the martensitic variant, improving the magnetic field-induced strain.

The introduction of texture in polycrystalline alloys can improve their superelasticity and plasticity. Ueland et al. [87] first proposed the concept of “Oligocrystalline structure” and defined “Oligocrystalline structure” as a grain morphology with a total surface area of the alloy that exceeds the total grain boundary area. The Cu-Zn-Al SMA was prepared into fibers and the bamboo-like oligocrystalline structure was obtained by heat treatment as follows: the wires drawn in the glass sheath were annealed in an argon atmosphere for 3 h at 800 °C and then quenched in water; they were subsequently kept at room temperature for at least 7 days. The fatigue life of the shape memory alloy was about 10^3^, which was two orders of magnitude higher than that of the bulk polycrystalline alloy.

Zhukov et al. [88,89,90,91,92] prepared a series of fibers of Ni-Mn-Ga, Ni-Mn-In, and Ni-Co-Mn-In alloys with a glass coating method and studied the microstructure, phase composition, and magnetic properties of the fibers, which enriched the research on the Ni-Mn-based “oligocrystalline structure” alloys. Zhang et al. [93] found that the maximum recoverable strain of superelasticity was 11.2% and the critical stress was 115 MPa in the Ni_54.0_Mn_24.1_Ga_21.9_ alloy fibers, as shown in Figure 4c. Glock et al. [94] studied the effect of annealing on the mechanical behavior and magnetic properties of a 7 M martensitic fiber. It was found that long-time annealing led to the growth of grains, the fiber changed from a polycrystalline to an oligocrystalline structure, and the reorientation stress of the twin variants decreased. Liu et al. [95] observed a large recoverable strain of 6.0% for Ni_44.5_Co_5.5_Mn_39.5_Sn_10.5_ fibers with a superelastic stress hysteresis of only 23 MPa. Figure 4d shows the stress–strain curves of the fibers at different stress levels. Ding et al. [96] reported that the Ni_52.8_Mn_23.8_Ga_23.4_ alloy fiber had a recoverable strain of 14.0%, as shown in Figure 4e. Li et al. [97] obtained 20% tensile recoverable strain in the <001>_A_ direction for the Ni_50.0_Mn_31.4_Sn_9.6_Fe_9.0_ alloy fiber, as shown in Figure 4b, which is the largest recoverable strain of Ni-Mn-based alloys reported to date.

In conclusion, if the free surface area caused by a reduction in material size is increased, the formation of an oligocrystalline structure can significantly increase the shape memory and superelastic strain recovery rate of the alloy and improve the fatigue life compared with that with an oligocrystalline structure produced by heat treatment.

**Figure 4 materials-16-05725-f004:**
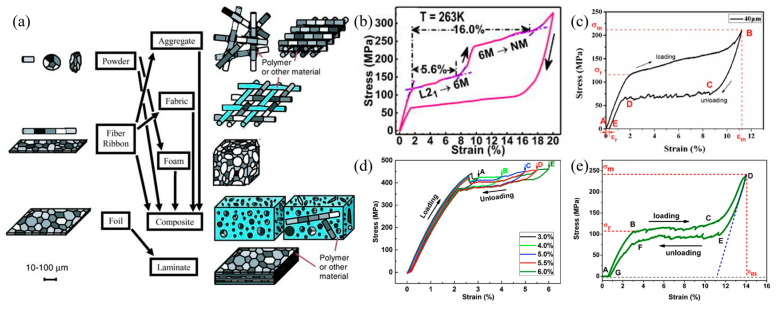
Toughening effect related to the sizes of Ni-Mn-based shape memory alloy. (**a**) Schematic illustration of the size effects of Ni-Mn-based ferromagnetic shape memory alloy (reproduced with permission from [86], John Wiley and Sons, 2011); (**b**) tensile stress–strain curve obtained at 263 K for the Ni_50.0_Mn_31.4_Sn_9.6_Fe_9.0_ microwire (reproduced with permission from [97], AIP Publishing, 2018); (**c**) tensile stress–strain curves of Ni_53.96_Mn_24.12_Ga_21.92_ wires with a diameter of 40 μm (reproduced with permission from [93], John Wiley and Sons, 2014); (**d**) tensile stress–strain curves of the Ni_44.5_Co_5.5_Mn_39.5_Sn_10.5_ microwire measured during different cycles with increasing strain levels at room temperature (reproduced with permission from [95], Elsevier, 2017); (**e**) tensile stress–strain curves obtained at 298 K for the Ni_52.87_Mn_23.82_Ga_23.32_ microwire (reproduced with permission from [96], IOP Publishing, 2017).

Ni-Mn-In thin ribbons prepared by the rapid cooling strip method had a uniform composition and well-developed columnar crystals and texture structure. Their brittleness was also improved to a certain extent. Feng et al. [98] prepared Ni_45_Mn_36.6_In_13.4_Co_5_ alloy ribbons with different thicknesses. The ribbons had a texture in the [001] crystallographic direction, and their fracture strength in this direction was improved. In addition, the texture of the ribbon was affected by the heat treatment method, and the texture was weakened by annealing. The higher the temperature, the more obviously the weakening phenomenon occurred. Hernando et al. [99] prepared Mn_50_Ni_40_In_10_ ribbons by the strip throwing process at a high cooling speed (∼10^4^–10^6^ Ks^−1^). The growth directions of austenite and martensite were [400] and [040], respectively, which were perpendicular to the strip surface. By applying magnetic fields in different directions, it was found that the magnetocaloric properties of the ribbons were improved in a specific direction. Inspired by the work of Hernando et al., further studies on the effects of texture on microstructure and mechanical properties were carried out. Akkera et al. [100] prepared Ni-Mn-In films with a thickness of 90–655 nm by magnetron sputtering and found that the grain size and crystallinity increased with increasing film thickness. Nanoindentation studies on the alloys showed that the film with a thickness of 153 nm had the highest hardness value (7.2 GPa) and an elastic modulus of 190 GPa.

It is worth noting that during the preparation of ribbons by rapid solidification, the melt crystallizes with a large degree of undercooling, resulting in a metastable structure. At the same time, substructures such as high-density dislocations are generated in the process, in which the stored deformation energy can act as driving forces of recrystallization. The substructures also affect the mechanical and functional properties of the alloy. Therefore, ribbon alloys prepared by rapid solidification process often need annealing treatment.

### 4.3. The Effects of d-Orbital Hybridization on the Toughness of Ni-Mn-Based Alloys

In 2015, Wei et al. [101,102] proposed Heusler-type all-*d*-metal alloys based on the concept of *d-d* hybridization. Unlike traditional Heusler alloys, which consist of two 3*d* transition metal elements (e.g., Ni, Mn) and one main group element (e.g., Ga, In, Sn), the all-*d*-metal alloys consisted of only 3*d*-metal elements. In these alloys, Ti was introduced to replace the main group element, and it was proven that *d-d* hybridization is beneficial to the stability of the alloy structure. Wei et al. [103] obtained 4.9% strain in the Ni_50_Mn_32_Ti_18_ alloy with a critical stress of 665 MPa. Yan et al. [17] prepared a Ni_50_Mn_31.75_Ti_18.25_ alloy with a directional solidification method. The alloy had a superelastic strain of 9% with almost no residual strain after unloading, and the fracture strain and tensile strength of the alloy were about 13% and 1.1 GPa, respectively. It is worth noting that the alloy produced an elastocaloric adiabatic temperature change of −20.4 K in the process of reverse martensitic transformation, which was larger than that of other Ni-Mn-based alloys [43,70,72,78,104] and compatible with that of Ni-Ti alloys [105]. In subsequent research, Yan et al. [19] designed a Ni_2_MnGa_1−x_Ti_x_ (x = 0, 0.25, 0.5, 1) alloy by partially replacing Ga with Ti. Theoretical calculations were performed on the basis of density functional theory (DFT), and it was found that the replacement of Ga with Ti could improve the toughness of Ni-Mn-Ga alloys. This is explained by the fact that the *d-d* orbital hybridization replaces the covalent *p-d* orbital hybridization, which reduces the covalent interaction between the constituent elements.

For the all-*d*-metal alloy, it was found that the relative volume of the alloy changed greatly during martensitic transformation, which was larger than that of other magnetic SMAs [106]. Therefore, it is expected that this alloy is highly sensitive to hydrostatic pressure and may have a high piezocaloric effect. However, so far, there are few studies on improving the mechanical and elastocaloric properties of Ni-Mn-based alloys with 3*d* orbital hybridization, and the relevant mechanism is still unexplored. For the Mn_50_Ni_30.5_Co_9.5_Ti_10_ alloy, since [−110]_cubic_/[100]_5M_ is the elongated direction during the transformation into 5M martensite, the resulting large strain could lead to a significant volume change of the alloy.

Shen et al. [107] systematically investigated the [001]-oriented Ni-Co-Mn-Ti polycrystals grown by directional solidification on the basis of the idea that the effects of element doping, preferential orientation, and *d*-orbital hybridization on the toughening of alloys can all be implemented in the alloy. The competition between the relatively weak *d-d* covalent hybridization and metallic bonds promotes the formation of multi-modulated structures, resulting in a low critical stress and a small stress hysteresis for superelastic deformation in the alloy. Li et al. [108] conducted a systematic study on Ni_2_Mn-based all-*d*-metal Heusler compounds by first principle calculations. They found that the compounds with early transition metals showed better ductility. Taking Ni_2_MnY and Ni_2_MnTa as examples, the nondirectional *d-d* interatomic hybridization prevailed, which facilitated metallic bonding in the alloys and consequently enhanced their ductility.

## 5. Summary and Outlook

In general, the practical application of Ni-Mn-based SMAs in solid-state refrigeration is restricted by their generic brittleness. In the past decades, the research on the toughening of Ni-Mn-based SMAs has advanced rapidly. Some toughening mechanisms, such as element doping, size effect, and *d-d* orbital hybridization, have been implemented in toughening of the Ni-Mn-based alloys and are summarized as follows:

(1) In Ni-Mn-based alloys with excessive doping of Cu, Fe, Co, and other transition metal elements, there are precipitates of γ second phase at the grain boundary, which are beneficial for improving the plasticity and toughness of the alloys. The improved toughness is related to the second phase, which could either hinder the initiation and propagation of micro-cracks or enhance the plastic deformation of the alloys. However, there is a negative effect in that the martensitic transformation may be hindered by the second phase, and the transformation hysteresis increases.

(2) The doping of rare earth elements, such as Gd, Tb, and Dy, can significantly refine the grain size, which is reduced with increasing contents of doping. Further increases in the addition content of rare earth elements lead to the formation of a rare earth metal-rich second phase in the Ni-Mn-based alloy matrix, where the transition from intergranular fracture to transgranular fracture occurs. However, when the rare earth element is excessively doped, the brittleness of the alloy increases. The doping of Y has similar effects.

(3) The doping of the B element can significantly refine the grains of Ni-Mn-based alloys because the “air mass” formed by B atoms can hinder the migration of grain boundaries.

(4) Ni-Mn-based alloys with textured structures can be manufactured by some advanced processing methods. The textured structures are effective in reducing stress concentration and inhibiting crack initiation in the alloys. Excellent cycle stability and large elastocaloric adiabatic temperature changes can be achieved in alloys with textured structures.

(5) Ni-Mn-based alloys can be prepared into samples with small dimensions, such as fibers, ribbons, and films. Through suitable heat treatment, the sample can contain grains with sizes compatible with its characteristic sizes, such as fiber diameter and ribbon (or film) thickness. The constraint and resistance effects of the grain boundary on the motions of the twin boundary are greatly reduced, improving the plasticity and superelastic strain recovery rates of the alloys.

(6) Ni-Mn-based alloys with *d-d* orbital hybridization have a reduction in brittleness compared with those with covalent *p-d* orbital hybridization, providing a new viable method for improving the toughness of the alloys.

Although plenty of studies on the toughening methods of Ni-Mn-based ferromagnetic shape memory alloys have been carried out, the toughness of the alloys is still far less satisfactory at their service conditions of tens of millions of mechanical cycles, as required by typical applications, such as elastocaloric refrigeration. Particularly, the toughening mechanisms of the alloys have to be further systematically investigated. It is suggested that future research should focus on the following three aspects:

(1) Studies on the microstructure–mechanical property relationship in Ni-Mn-based ferromagnetic shape memory alloys are very necessary. The clarification of such a relationship is of significance in that the shape memory alloys with superior mechanical properties can be developed by designing the specific microstructures and processing parameters of heat treatment. 

(2) It is desirable that the mechanisms of failure and enhancement in toughness related to phase transformations in Ni-Mn-based ferromagnetic shape memory alloys are investigated. Since shape memory alloys undergo multi-cycle phase transformations under large applied stresses, the energy loss and defect accumulation during the phase transformation processes are key factors that could affect the failure of shape memory alloys. It is very important to study the relationships between high-cycle phase transformations and failure mechanisms in Ni-Mn-based alloys, reasonably regulating their service lifetime and cyclic stability.

(3) Ni-Mn-based ferromagnetic shape memory alloys could exhibit magnetic and structural phase transformations, and they have the great advantage of multi-caloric effects. More importantly, we can make full use of those multi-caloric effects, such as elastocaloric, magnetocaloric, and piezocaloric effects of high-strength and toughened Ni-Mn-based alloys, facilitating the practical application of Ni-Mn-based ferromagnetic shape memory alloys in the field of solid-state refrigeration.

## Figures and Tables

**Figure 3 materials-16-05725-f003:**
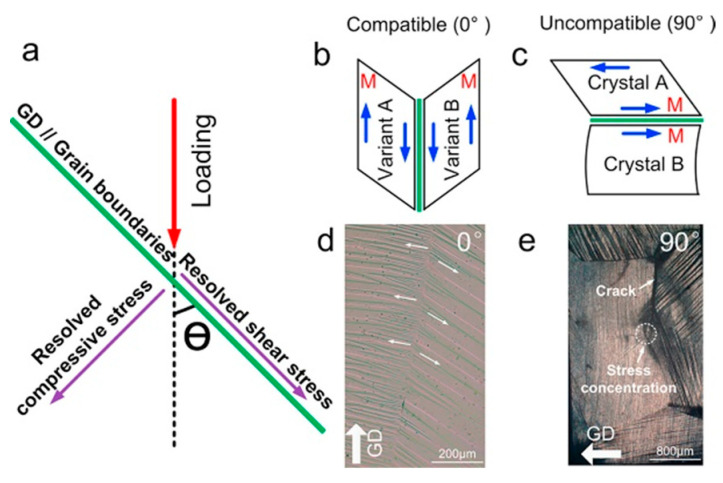
(**a**) A schematic of the stress distribution in the stress field surrounding the grain boundaries (green lines). (**b**,**c**) Schematics of the relationship between the compatibility at grain boundaries and orientation angle. The red letter “M” stands for the martensitic phase. (**d**,**e**) Optical images of surface topography for the compressed samples (reproduced with permission from [73], Elsevier, 2014).

**Table 1 materials-16-05725-t001:** Toughening methods, mechanism, and effects for Ni-Mn-based shape memory alloys.

Toughening Method	Toughening Mechanism	Toughening Effect	Reference
Ni-Mn-Ga alloy doped with Cu	Solid solution strengthening and toughening	Shape memory effect of 6.2%, compressive plasticity of 22%, and compressive strength of 878 MPa	[27]
Ni-Mn-Ga alloy doped with Cu	Second phase strengthening and toughening	Compressive strain > 70%, maximum compressive strength of 785.6 MPa, tensile strain of 6.2%	[28]
Ni-Mn-Sn alloy doped with Cu	Second phase strengthening and toughening	Breaking strength of 1150 MPa	[29]
Ni-Mn-Ga single crystal alloy doped with Fe	Solid solution strengthening and toughening	Brittleness of the alloy is obviously improved, and the Vickers hardness is 6.4 GPa	[30]
Ni-Mn-Ga alloy doped with Fe	Solid solution and second phase strengthening and toughening	Transgranular fracture, fracture toughness of 18 N/mm^3/2^	[32]
Ni-Mn-In alloy doped with Fe	Solid solution and second phase strengthening and toughening	Compressive strength of 1200 MPa, maximum compressive strain of 15.8%	[33]
Ni-Mn-In alloy doped with Fe	Second phase strengthening and toughening	More than 16,000 mechanical cycles under 300 MPa without structural degradation	[34]
Ni-Mn-Ga alloy doped with Co	Second phase strengthening and toughening	Tensile strength of 729 MPa, breaking strain of 14.1%	[42]
Ni-Mn-Ga alloy doped with Cr and Co	Second phase strengthening and toughening	Ni_45_Mn_36_In_13_Co_5_Cr alloy breaking strain of 5%, Ni_45_Mn_35_In_13_Co_5_Cr_2_ alloy breaking strength of 550 MPa	[43]
Ni-Mn-Ga alloy doped with Gd	Grain refinement strengthening and toughening	Shape memory recovery rate of Gd1 alloy is 100%, and the maximum shape memory strain is 1.9%	[45,46]
Ni-Mn-Sn alloy and Ni-Mn-In alloy doped with Gd	Second phase strengthening and toughening, changing the fracture mode of alloy	Compressive strength increased from 448 MPa to 707 MPa, the compressive strain increased from 4.5% to 9.0%	[48]
Ni-Co-Mn-In alloy doped with Gd	Second phase strengthening and toughening, distorting the lattice	Compressive elongation is 8.8%; compressive strength is 992 MPa, 5.5 times that of undoped alloy	[49]
Ni-Mn-Sn alloy doped with Tb	Grain refinement and second phase strengthening and toughening	Compressive strength of Tb2 alloy is 571.8 MPa, and the fracture strain is 22.0%, which is higher than the 74.3 MPa and 9.2% of the undoped alloy	[50]
Ni-Mn-In alloy and Ni-Mn-Ga alloy doped with Tb	Second phase strengthening and toughening, changing the fracture mode of alloy	Tb0.4 alloy can withstand 622 MPa uniaxial stress; Tb1 alloy shape memory strain of 2.68%	[51]
Ni-Mn-Ga alloy doped with Tb	Grain refinement and increasing grain boundary strength	Tb0.3 alloy compressive strength of 780 MPa	[53]
Ni-Mn-Ga alloy doped with Nd	Enhancing the bonding force of the grain boundary	Fracture strain of 6%	[54]
Ni-Mn-Ga alloy doped with Dy	Changing the fracture mode of alloy	Dy1 alloy compressive strain reaches its maximum	[55,56,57]
Ni-Mn-Ga alloy doped with Dy	Forming granular precipitates	Compressive strength of ~855.9 MPa, fracture strain of ~12.7%	[58]
Ni-Mn-Ga alloy doped with Y	Second phase strengthening and toughening, changing the fracture mode of alloy	The type of fracture changes from intergranular fracture to transgranular fracture	[60]
Ni-Mn-In-Co alloy doped with B	Modifying alloy grain boundary, second phase strengthening and toughening	Form a structure similar to sub-grain, the average size of the second phase is 1–2 μm	[62]
Ni-Mn-In alloy and Ni-Mn-In-Fe alloy doped with B	Enhancing the bonding force of the grain boundary by microalloying, grain refinement	No decay of the elastocaloric effect after 150 cycles	[66]
Ni-Mn-In-Cu alloy doped with B	Second phase strengthening and toughening	Stable for 100 cycles at 3% strain	[4]
Ni-Co-Mn-Sn alloy doped with Al	Reducing stacking fault energy	Compressive fracture strain of 8.7%, compressive strength of 1050 MPa	[68]

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
