# Peer review of "Toughening of Ni-Mn-Based Polycrystalline Ferromagnetic Shape Memory Alloys"

_materials, 2023, doi:10.3390/ma16165725_

Round 1
Reviewer 1 Report
This is a good work. I enjoyed reading it. Major revisions are in order.
Language needs to be improved. Several errors found.
“on the toughening methods and toughening mechanisms of Ni-Mn-based alloys, while a systematic summary on the progresses in these studies is lacking”: and why is that? Detail.
“The doping of ferromagnetic element Fe in Ni-Mn-Ga alloys can improve the magnetic properties of the alloys, which has attracted much attentio”: always? Not really.
“the fracture of alloys indicated that the Ni48.7Mn30.1Ga21.2 (x=0) alloy had an intergranular fracture with a fracture toughness of”: what drove this? Discuss.
“Y doping improves the yield strength and plasticity of the alloy, and the fracture type changes from intergranular fracture to transgranular cleavage fracture.”: but does it change in the microstructure that drives this?
“of Ni-Mn-based alloys can be improved by doping B element”: how much B is of interest?
Fig 2 d) cannot be seen.
“The material’s characteristic sizes at micro- and nano-scales, such as the grain sizes or its dimensions, can significantly affect its mechanical behaviors”: true for any engineering alloy. See 10.1016/j.msea.2023.144946 and 10.1016/j.msea.2023.144722 and complement.
“d the bamboo-like oligocrystalline structure is obtained by heat”: what heat treatment schedule? Must be detailed.
“e relative volume of the alloy changes greatly during martensitic transformatio”: and why? Discuss.
Minor changes needed
Reviewer 2 Report
The review is well organized and supported by sufficient on numbers of references. Nonetheless, some minor revisions are necessary:
i) a use of terms “austenite” and “martensite” needs more detail description. There is only not very precise notice (rows 80-82) that concerns the transformation of crystal structure = mentioned 10M and 14M are not “structures”, they are modulations of tetragonal structure. Moreover, the most important martensitic (specific) transformation from cubic to tetragonal (or orthorhombic) crystal structure that is a background of all the shape memory, elasto- and baro-caloric effects in the studied materials is not mentioned at all!
ii) In the sentence (rows 98-99) “The formation of chemical bonds leads to the reduction of atomic binding force at the grain boundary”, the chemical bonds should be specified = bonds with impurities, substituted atoms?
iii) Fig. 2d) is not lucid without magnification. The unit N/mm3/2 is wrong (row 152, 153), N/mm2 = MPa. Element Y (row 614) don’t belong to rare earth elements.
iv) There are mistakes in sentence “Later, Feng et al. [30] replaced Mn…” (row 145), i.e., Feng is not author of paper in ref. [30] and citation of paper of F. Xue in ref. [30] is wrong, as well as citation of paper [31].
v) Data concerning page or number of papers are missing in many citations: [2,4,6,11,14,15,17-19,21,34,38,39,50,58.74,76,75-83,89,94,95,97,99-101,104 and 106]
vi) There are many misprints or solecisms in the text, see, e.g., rows: 47,49,64,77,93,100, 220,363,569,570,575,629,647. Don’t use comma in front of “and”. It should be probably better to use word “proportions” instead of “sizes” in “…characteristic sizes at micro…” (row 484). The “imperative” (rows 657-660) should be formulate more understandably.
There are many misprints or solecisms in the text, see, e.g., rows: 47,49,64,77,93,100, 220,363,569,570,575,629,647. Don’t use comma in front of “and”. It should be probably better to use word “proportions” instead of “sizes” in “…characteristic sizes at micro…” (row 484). The “imperative” (rows 657-660) should be formulate more understandably.
Reviewer 3 Report
The paper Toughening of Ni-Mn-Based Polycrystalline Ferromagnetic Shape Memory Alloys written by Siyao Ma, Xuexi Zhang, Guangping Zheng, Mingfang Qian and Lin Geng is written well and is consistent in information about Ni-Mn SMAs. This study significantly contributes to expanding the understanding research directions of Ni-Mn-based ferromagnetic SMAs.
See following some minor observations:
Line 77: (1/2, 1) /2, 1/2)
Line 82: explain NM and M from 10M, respectively, 14M (NM- non-modulated, M- refers to the monoclinicity)
Line 96: insert reference
3.3. Section: Figure 2 (b), (c) and (d) are not explained in the text. The authors start with Figure 2 (f). Please remediate. Perhaps it would be more useful if figure 2 were divided into two distinct figures: mechanical properties (Fig. 2 (e, f, g) and structural (the rest).
Line 417. I suggest replace Ni52Mn32In16 with NiMnIn.
Use Figure or Fig. … in all article.
Round 2
Reviewer 1 Report
All comments fully addressed. Acceptance is recommended.
ok